# Bad Exoplanet! Explaining Degraded Performance when Reconstructing Exoplanets Atmospheric Parameters

**Alkis Koudounas**[*]
Politecnico di Torino
Turin, Italy

**Flavio Giobergia**
Politecnico di Torino
Turin, Italy

**Elena Baralis**
Politecnico di Torino
Turin, Italy

## Abstract

Deep learning techniques have been widely adopted to automate the reconstruction of atmospheric parameters in exoplanets, at a fraction of the computational cost required by traditional approaches. However, many of the reconstruction models used are intrinsically non-interpretable. With this work, we aim to produce descriptions for the characteristics of exoplanets that make their atmospheric composition reconstruction problematic. We present a model-agnostic approach to detect biased data subgroups described via atmospheric parameters such as planet distance and surface gravity.

We show that adopting an ensemble approach remarkably improves the quality of the outcomes overall, as well as at the subgroup level, on synthetic data simulated for the upcoming Ariel space mission. Experimental results further demonstrate the effectiveness of adopting explanation techniques in identifying and describing significant performance gaps between weak learners and their ensemble. Our work provides a more nuanced description of the results provided by deep learning techniques, to enable more meaningful assessments of what can be reasonably achieved with them.

## 1 Introduction

Exoplanets are planets that orbit stars outside of the solar system. Their discovery has profoundly transformed our understanding of planetary science. Characterizing the atmospheric diversity observed across these worlds is crucial for contextualizing our own solar system, the planets within it, and what exists outside of it. The most common technique is transmission spectroscopy, which measures the absorption signatures in a planet's atmosphere during a transit event (1). This reveals properties like chemical composition, temperature, presence, and characteristics of clouds.

Over the past two decades, transmission spectra have been obtained for a few dozen exoplanets. However, upcoming missions promise to increase the quantity and quality of this data vastly. The James Webb Space Telescope will soon more than triple existing observations (2), while the Ariel space mission (3) aims to survey over 1,000 exoplanet atmospheres.

Atmospheric retrieval from low-resolution transmission spectra poses substantial difficulties due to the complex, high-dimensional parameter space involved in modeling exoplanet atmospheres. This introduces degeneracies wherein multiple solutions could plausibly fit the observed spectral features (4). Rather than a single definitive solution, a posterior probability distribution is necessary to disentangle these degenerate cases. This distribution details the range of atmospheric compositions and properties compatible with the measurements and associated uncertainties, thereby providing a more informative characterization of the planet's atmosphere than point estimates alone.

Traditional Bayesian retrieval methods employing computationally expensive sampling algorithms

---

[*]Corresponding Author: `alkis.koudounas@polito.it`

NeurIPS 2023 AI for Science Workshop.

currently offer the best results (5). However, when applying these methods at scale to massive upcoming datasets like those from Ariel, the required runtimes become impractical (6).

To address this computational challenge, machine learning (ML) and deep learning (DL) techniques have been applied in exoplanetary research. These methods have already been used in data processing tasks, including data detrending (7; 8), debris removal (9; 10), and planet detection and characterization (11; 12; 13). Various ML and DL approaches, such as random forests, convolutional neural networks (CNNs), and generative adversarial networks (GANs), have been employed to improve retrieval efficiency and reduce computational time. However, these approaches often involve approximations that can affect the accuracy of posterior estimations. A recent study (14) proposes a multimodal 1D-CNN that combines spectral data with auxiliary data related to stellar and planetary parameters. The authors demonstrate that this approach yields favorable outcomes in approximating the posterior distribution, all while exhibiting efficient computational performance.

Though machine learning models have demonstrated faster reconstruction capabilities, interpretability remains a limitation of many existing solutions. In particular, models may systematically struggle with the reconstruction of specific categories of exoplanets. However, when reporting aggregated results, these patterns generally do not emerge. As such, our goal is to develop an interpretable framework for probabilistic atmospheric characterization that provides human-interpretable descriptions of the patterns for which machine learning algorithms struggle to build accurate predictions.

Exoplanet transmission spectra are often accompanied by additional planetary attributes such as mass, surface gravity, stellar proximity, etc. We call this auxiliary information *planetary* metadata. Combinations of these metadata values delineate distinct data *subgroups*. Typically, model performance is evaluated either globally or on predefined important subgroups.

We introduce efficient techniques to comprehensively compare model performance across all metadata-induced subgroups without an *a priori* definition of the relevant subgroups. Naively enumerating and evaluating every possible subgroup is infeasible due to their combinatorial scaling with metadata dimensionality. Instead, we adapt methods from bias analysis to focus only on "frequent" subgroups accounting for a meaningful fraction of data (e.g., 0.5% of the dataset), whose number does not grow at the same rate as the total number of possible subgroups. Our approach allows measuring and contrasting model performance within statistically and practically significant frequent subgroups. This offers insights beyond traditional evaluation by disclosing disparities across subgroups that may average out or remain hidden at the overall, aggregate level. The identification of subgroups where a model succeeds or fails can guide the development of more robust solutions.

In recent research, several studies (15; 16; 17) have focused on automatically identifying subgroups with problematic behaviors in structured data. Our work draws inspiration from DIVEXPLORER (15) and introduces a method for comparing models within these subgroups. While other heuristic-driven exploration approaches do not enable model comparison (16; 17), DIVEXPLORER stands out as the only method capable of supporting this comparison due to its comprehensive exploration of frequent subgroups, as already shown in (18). To the best of our knowledge, the only prior study that delves into the interpretation of such deep learning models for exoplanet atmospheric retrievals is (19). In their work, the authors primarily investigate and quantify how the predictions of one atmospheric parameter change conditioned on the variation of another. Their method is not tied to any specific model, just like our approach. In contrast, our analysis goes a step further by examining the model's performance at a subgroup level, taking into account *all* possible recurring subgroups created by combining different metadata criteria. Unlike (19), we also explore how auxiliary stellar and planetary parameters and their combinations influence the model predictions.

Through experiments using synthetic observations simulated for the Ariel mission, we showcase the preliminary results that demonstrate the potential for our proposed framework to characterize the patterns that lead to degraded performance in the reconstruction of exoplanet atmospheres. We argue that interpretable machine learning techniques offer a promising approach for gaining scientific understanding from upcoming large astronomy surveys and advancing the field of exoplanet research.

The remainder of the paper is organized as follows: Section 2 reviews the problem under analysis, along with an exploration of the data and target task. Section 3 introduces the relevant aspects of the proposed methodology for the identification of divergent subgroups. Section 4 reports the main experimental results obtained. Finally, Section 5 draws conclusions and possible future directions.

## 2 Exoplanets Atmosphere Reconstruction

Exoplanets are typically discovered using methods like radial velocity and transit observations (20). When an exoplanet transits in front of its host star, it causes a slight decrease in the star's brightness as seen from Earth. During a transit, the exoplanet's atmosphere influences the entity of the dip in the star's brightness in a wavelength-dependent fashion through the absorption of specific wavelengths. This reveals details about the atmosphere's composition and properties. Astronomers can analyze the star's apparent dimming at different wavelengths, known as transit *spectra*, to indirectly deduce characteristics of the atmosphere. This process of inferring atmospheric information from transit spectra is called the inverse problem (21). However, noise within the spectral data and limited wavelength coverage can result in information loss and uncertainty in atmospheric models.

The goal of atmospheric retrieval is to estimate parameters that best fit the observed spectrum using a forward model and optimizer. In a Bayesian framework, it aims to determine the posterior distribution, representing the likelihood of model parameters given the observed data.

### 2.1 Data and metric

In our experiments, we use the data provided by the Ariel Big Challenge Database (22): it consists of spectral data, which quantifies the absorption of light by the exoplanet's atmosphere at various wavelengths. This can be used as a proxy for the atmospheric composition since different gases have different signatures in terms of light absorbed.

The task is to predict the combined distribution of six atmospheric properties, namely the temperature and the log-abundance of five gases, based on the observed spectrum. This prediction is represented as the posterior distribution. To assess the quality of the predictions, we employ the 2 Sample Kolmogorov–Smirnov test (K-S test). The K-S test is a well-established statistical method that determines whether two given samples come from the same underlying continuous distribution. Its score ranges from 0 to 1, with lower scores indicating higher similarity between the samples and a score of 0 indicating perfect similarity. In particular, we compute the K-S statistic for each planet and compute the overall performance of a model as the average K-S statistic across a separate test set.

## 3 Methodology

This section introduces the main concepts of interest for the proposed framework. Our main goal is to introduce a model-agnostic way of describing the main patterns that lead to degraded performance in the reconstruction of atmospheric parameters. As such, this section first introduces one possible deep learning-based approach that has already been used in literature to solve the exact task of interest. Then, we present how the patterns (in terms of recurring subgroups) can be extracted and analyzed.

### 3.1 Atmospheric parameters estimation

Without loss of generality for the explanation technique adopted, we use, as the main model under study, the one proposed in (14). The approach leverages both the spectral data and planetary metadata available to produce the mean vector and the full covariance matrix for the distribution of the target atmospheric parameters – with the underlying assumption that the distribution of parameters follows a normal distribution. To this end, the spectral data is processed through a convolutional model, whereas the auxiliary data is directly fed to a feed-forward model.

As a way to reduce the variance of the approach proposed in (14), we adopt an ensemble of learners trained on random subsets of the training data. We adopted as the ensemble prediction the average prediction over all of the learners, both for the predicted means and covariance matrices. While this approach does not account for different confidences in the predictions of the weak learners (i.e., all models are assigned equal weight in the final result), we empirically show that the adoption of a modest number of models still significantly outperforms the performance of the single model.

### 3.2 Subgroups identification

We define data subgroups using itemsets, which are sets of *attribute-value* pairs. For the models under analysis, the *divergence* of a data subgroup refers to the difference between the model's performance on that subgroup and its performance on the entire dataset. Similarly, the *subgroup gain* is the

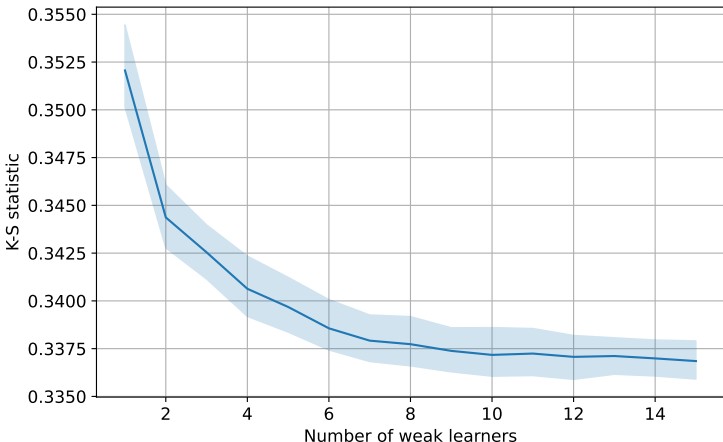

Figure 1: K-S statistic computed as the number of estimators in an ensemble increases. The estimated parameters are obtained as the average across all learners. Error bars are obtained as the 95% confidence interval when running the experiment 10 times, with different initializations and train/test splits.

difference in performance between two models on a specific subgroup.

To annotate the simulated data from Ariel, we leverage the planetary metadata available, which consists of eight interpretable attributes describing planet- and star-dependent parameters, including star distance, stellar mass, radius and temperature, planet mass, orbital period, semi-major axis, and surface gravity. Since the available metadata is continuous in nature, for this preliminary work, we discretize its values using 10 equal-frequency bins for each attribute. This decision affects the granularity with which the subgroups can be described and might be subject to changes based on domain expertise.

**Items and itemsets.** Let $D$ be our dataset, $\mathcal{A}$ its set of metadata attributes, and $\mathcal{I}$ its set of *items*. Each item takes the form $a = v$, where $a$ represents an attribute $\in \mathcal{A}$ and $v$ indicates the corresponding value. Since $v$ represents a bin of values, it can be represented as an interval $[x, y)$: as such, an alternative representation for the item is $a \in [x, y)$. For instance, if we consider attributes like *planet surface gravity* and *planet distance*, examples of items could be {*planet_surface_gravity* $\in [13.10, 18.38)$ $ms^{-2}$} and {*star_radius* $\in [1.62, 6.30)$ $R_\odot$}.

The *subgroup* associated with a particular item is the portion of the dataset that meets the criteria set by that item. For each attribute, we require that the subgroups created by items should form a partition of the dataset. For instance, when it comes to the *star radius* attribute, the defined ranges should not overlap, and their combination should encompass all possible radii within a predefined set. An *item* allows us to effectively *slice* or select a subset of the data based on a single attribute. Additionally, we can perform multi-attribute slicing by considering *itemsets*, which are collections of zero or more items. Each item within an itemset pertains to a distinct attribute. For example, an *itemset* might look like {*planet_mass* $\in [0.01, 0.02)$ $M_J$, *star_radius* $\in [1.62, 6.30)$ $R_\odot$}.

The *support* of an itemset $I$ is defined as the fraction of the dataset that corresponds to $I$, that is, the ratio of the size of the subgroup that satisfies the criteria specified by $I$ to the size of the entire dataset. Consequently, an itemset with a support of 0.01 indicates that it appears in 1% of the dataset. The empty itemset, which represents the entire dataset, has a support of 1.

**Subgroup divergence and gain.** Let's define a performance measure $f$ for a particular task (e.g., the K-S statistic). For a given model $M$ and a subgroup (referred to as an itemset) denoted as $S$, we use $f(S, M)$ to represent the performance of the model on that specific subgroup. The *divergence* related to a subgroup $S$ with respect to model $M$ indicates the disparity between the model's performance on the subgroup $S$ and its performance on the entire dataset as a whole (15):

$$div_f(S, M) = f(S, M) - f(\emptyset, M) \, . \tag{1}$$

| Subgroup | score | $\Delta_{score}$ | t |
|---|---|---|---|
| {*planet_orbital_period* ∈ *[34.76, 731.94) days*, *planet_surface_gravity* ∈ *[13.10, 18.38) ms⁻²*} | 0.413 | 0.076 | 3.276 |
| {*planet_semi_major_axis* ∈ *[0.01, 0.04) AU*, *planet_surface_gravity* ∈ *[4.36, 5.58) ms⁻²*} | 0.412 | 0.075 | 2.808 |
| {*planet_semi_major_axis* ∈ *[0.21, 1.50) AU*, *planet_surface_gravity* ∈ *[13.10, 18.38) ms⁻²*} | 0.409 | 0.072 | 3.256 |

Table 1: Top-3 highest negatively divergent subgroups on performance for the *ensemble*. The *t* column indicates the Welch's t-test.

We establish the concept of *gain* when transitioning from model $M_1$ to model $M_2$ for a specific subgroup $S$. This *gain* refers to the improvement in performance achieved on the itemset $S$ when model $M_1$ is substituted with model $M_2$:

$$gain_f(S, M_1, M_2) = f(S, M_2) - f(S, M_1) \ . \tag{2}$$

We utilize DIVEXPLORER (15) to detect itemsets with significant divergence or gain values in terms of their absolute values. DIVEXPLORER employs efficient frequent pattern mining methods to extract all itemsets that surpass a specified support threshold, along with their corresponding divergence values. The support threshold plays a crucial role in this process as it serves to control the exploration. It ensures that the itemsets returned by the tool contain a sufficient amount of data to be considered both statistically and operationally meaningful.

**Shapley values.** We are interested in understanding the role of individual items in generating subgroups that exhibit significant divergence or gain, where $g$ represents the metric of interest, which could be either divergence or gain. Building on the principles outlined in (15), we introduce the concept of an item's contribution to the metric $g(S)$ for subgroups.

To calculate this contribution, we draw upon the game theory-based idea of the *Shapley value*. In this framework, each item $i$ within the subgroup (i.e., itemset) $S$ is attributed a contribution value based on how it influences the divergence or gain of the entire itemset. This value is denoted as $s_g(i, S)$, representing the extent to which $i$ contributes to the divergence or gain of $S$. Importantly, the sum of the Shapley values for all items within $S$ equals the overall divergence or gain of the subgroup, i.e., $\sum_{i \in S} s_g(i, S) = g(S)$. Additionally, we examine the *global* Shapley value, referred to as $\mathcal{S}_g(i)$ for an individual item $i$. This value measures the average impact of including item $i$ in all possible compatible subgroups (15).

## 4 Experimental results

We assess the effectiveness of our approach through various means, including its capability to uncover error sources (§4.1), a comparative analysis of subgroup behavior across different models (§4.2), and an examination of the global role played by attributes (§4.3). Our code to reproduce the experiments is accessible at the following link: `https://github.com/koudounasalkis/Ex-o-plain`.

### 4.1 Individual Model Analysis

We begin by showing that the adoption of an ensemble of models results in a meaningful improvement in performance with respect to individual models, as measured through a K-S test, over various runs. Figure 1 highlights this aspect, showing how a number of learners of approximately 5 leads to a reasonable trade-off between computational power required and quality of the obtained performance. Based on this consideration, the rest of the experimental section will focus on the study of an ensemble of 5 learners and the comparisons that can be made between the ensemble and a single weak learner.

Next, we investigate the origins of errors made by the ensemble model. In Table 1, we present information about the top 3 itemsets with the most significant negative divergence. This analysis is conducted with a support threshold of 0.01 and a redundancy threshold of 0.03 to ensure that the retrieved subgroups are not completely overlapped, as detailed in (15). These particular itemsets represent subgroups where the model's performance falls below the average performance level.

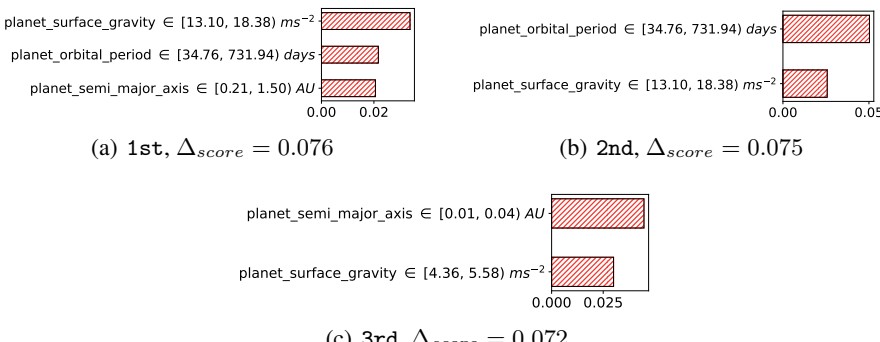

(a) 1st, $\Delta_{score} = 0.076$    (b) 2nd, $\Delta_{score} = 0.075$

(c) 3rd, $\Delta_{score} = 0.072$

Figure 2: ENSEMBLE. Item contribution to the final score for (a) the subgroup with the highest negative divergence (the lower the score, the better), (b) the second, and (c) the third.

| | Subgroups | $gain_{score}$ | individual *score* | ensemble *score* |
|---|---|---|---|---|
| ↑ | {*star_radius* ∈ *[1.62, 6.30)* $R_\odot$, *planet_surface_gravity* ∈ *[18.38, 244.88)* $ms^{-2}$} | -0.041 | 0.381 | 0.340 |
| = | {*star_radius* ∈ *[0.30, 0.72)* $R_\odot$, *planet_mass* ∈ *[0.01, 0.02)* $M_J$} | 0.00 | 0.323 | 0.323 |
| ↓ | {*star_radius* ∈ *[0.72, 0.81)* $R_\odot$, *planet_semi_major_axis* ∈ *[0.04, 0.05)* $AU$} | 0.017 | 0.327 | 0.344 |

Table 2: Performance gain when transitioning from *individual* to *ensemble* models on itemsets where performance increases the most (↑), decreases (↓), or remains equal (=).

Interestingly, we find that planet surface gravity is a significant factor in our analysis. It appears in all three of the most negatively divergent subgroups, with varying values falling within the ranges of either $13.10$ to $18.38$ $ms^{-2}$ or $4.36$ to $5.58$ $ms^{-2}$.

The most substantial drop in performance, amounting to a decrease of 0.076, is observed within the subgroup defined by the planet's orbital period in the range of 34.76 to 731.94 days and the planet's surface gravity in the range of $13.10$ to $18.38$ $ms^{-2}$.

When examining a subgroup, it becomes important to further understand the significance of each metadata attribute in either contributing to or diminishing the divergence within that subgroup. This significance is quantified through the concept of Shapley value, which is defined in Section 3. In Fig.2, we can observe the individual impact of each attribute within the three most negatively divergent subgroups, as listed in Table1. The analysis reveals that within the first subgroup (Fig.2(a)), a high planet orbital period carries more weight than surface gravity in terms of influence. In the second and third subgroups (Figs.2(b) and 2(c)), the planet's semi-major axis wields a greater influence compared to surface gravity.

## 4.2 Models Comparison Analysis

We conduct a comparison between the performance of an individual weak learner and that of the ensemble model. The overall score shows an improvement, rising from 0.347 for the individual model to 0.336 for the ensemble model. We recall that the lower the score, the better the model. This increase in score further suggests that the ensemble indeed surpasses the performance of the individual learners. To further analyze subgroup performance, we calculate the *gain* as the performance difference between the individual and ensemble models within each subgroup. Our analysis revealed that performance improved in 91.83% of the explored subgroups, while it decreased in 8.16% of them. Figure 3 shows the gain distribution when transitioning from the individual weak learner to the ensemble model, with cross-hatched green indicating subgroups showing performance improvement and red a decrease. Table 2 highlights the subgroups in which the performance increases the most (↑), remains relatively unchanged (=), or decreases (↓) when transitioning from the individual to the ensemble model. The larger improvement in performance (-0.041) corresponds to the subgroup characterized by {*planet_surface_gravity* ∈ *[18.38, 244.88)* $ms^{-2}$, *star_radius* ∈ *[1.62, 6.30)* $R_\odot$}.

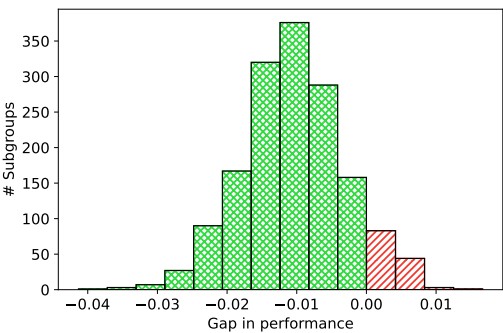

Figure 3: Distribution of gain ensemble-individual. Cross-hatched green denotes performance improvement when going from the individual model to the ensemble (the lower the score, the better), while red indicates performance decrease.

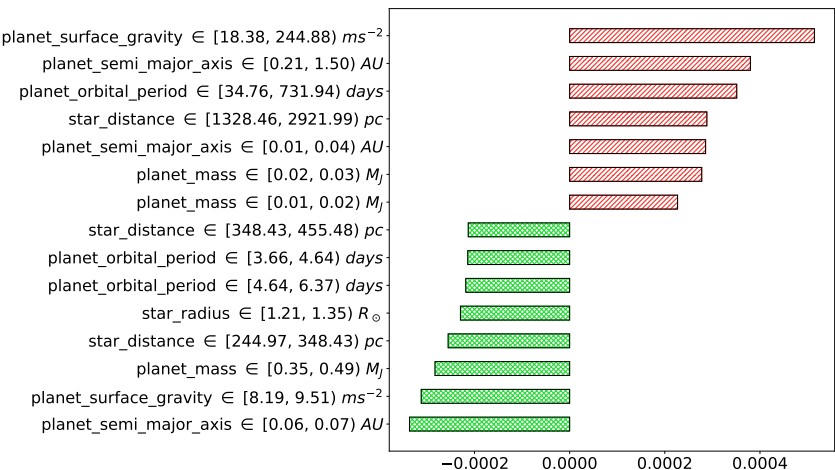

Figure 4: Global Shapley values of the ensemble.

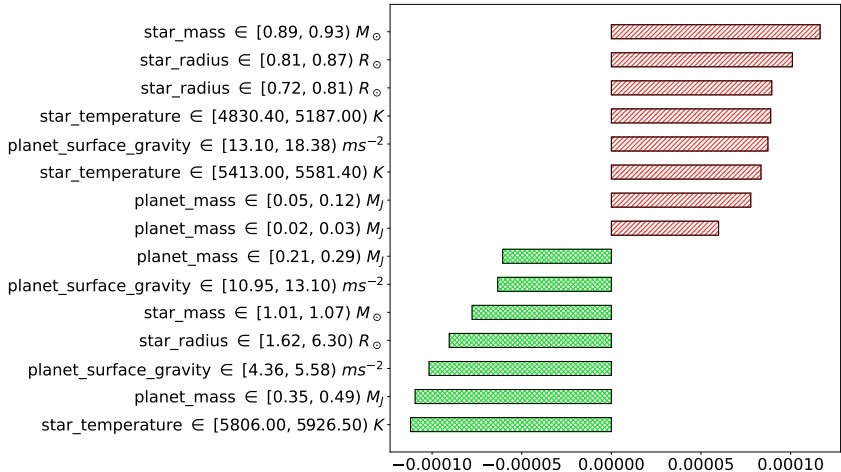

Figure 5: Global Shapley values of the gain when transitioning from individual to ensemble.

### 4.3 Global Divergence

We provide a summary of the impact of each item on two aspects: the performance of the ensemble model and the performance improvement observed when transitioning from the individual learner to the ensemble. This evaluation is conducted using the global Shapley value, denoted as $\mathcal{S}_g$. In essence, a positive value for $\mathcal{S}_g(i)$ signifies that, on average, including item $i$ in a subgroup $S$ (where $i \notin S$) results in an enhancement in performance. Conversely, negative values indicate the opposite.

The top 15 items with the most significant impact on the ensemble model's performance are depicted in Figure 4. The highest performance improvement is observed for a range of small values for *planet_semi_major_axis* (i.e., $\in [0.06, 0.07)\,AU$) and relatively small for *planet_surface_gravity* (i.e., $[8.19, 9.51)\,ms^{-2}$). Conversely, higher values for both these attributes result in a performance drop.

Additionally, Figure 5 displays the top 15 items with the most substantial impact when transitioning from the weak learner to the ensemble. In this scenario, we observe that the most significant improvements and declines in performance are associated with parameters related to the host star. In the case of the highest performance increase, this is linked to high values of star temperature falling within the range of $5806.00$ to $5926.50\,K$.

Conversely, the most substantial performance decrease is connected to stars with relatively small mass $\in [0.89, 0.93)\,M_\odot$, and small star radii, either $\in [0.81, 0.87)\,R_\odot$ or $\in [0.72, 0.81)\,R_\odot$.

## 5  Conclusions

In this work, we explored the adoption of interpretability techniques to offer descriptions of the situations of degraded performance that can occur when estimating atmospheric parameters of exoplanets. We do so by identifying frequent subgroups (as defined by auxiliary data regarding the star/planet system) for which a significant variation in performance can be observed.

Although the proposed results concern synthetic datasets, we argue that the same technique can and should be used to provide better insights regarding the contexts that can produce degraded (or improved) reconstructions. This is helpful both as a way of weighting the results obtained (as well as their validity) and for identifying situations in the current experimental design that could benefit from additional work.

We additionally show that ensembles of learners, as is well known in the literature, produce overall better performance when compared to single weak learners. We further provide descriptions of the most relevant changes in subgroups' performance that justify this change.

We plan on further expanding the process of identifying degraded performance so as to produce more meaningful descriptions that do not rely on the discretization of the auxiliary information available. We are additionally interested in improving the performance of the learners adopted by, for example, leveraging loss functions that better reflect the nature of the parameters being estimated.

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

# A  Appendix A

On top of describing the situations of degraded performance, our approach also allows us to identify subgroups where the model (i.e., the ensemble in our case) performs significantly better (w.r.t. the overall performance). Table 3 shows the top three subgroups with the most improved scores. Notably, these subgroups contain characteristics of both the host star and planet. The largest improvement in performance (-0.055) is seen in the subgroup defined by {*star_radius* $\in [1.21, 1.35)$ $R_{\odot}$, *planet_semi_major_axis* $\in [0.04, 0.05)$ $AU$}.

Interestingly, the item *planet_semi_major_axis* $\in [0.04, 0.05)$ $AU$ also defines one of the subgroups with the most increased error, as shown in Table 1. Analyzing the Shapley values for each variable's contribution to subgroup divergence (Fig. 6) reveals that the host star radius has a greater impact on this subgroup than the planet's semi-major axis (Fig. 6(a)). Similarly, for the second-best subgroup, the stellar properties are more influential than planetary ones (Fig. 6(b)), while the opposite holds true for the third subgroup (Fig. 6(c)).

# B  Appendix B

As already shown in the main document, we calculated the *gain* as the difference in prediction error between the individual models and the ensemble model within each subgroup. Our analysis found performance improved in 91.83% of subgroups while decreasing in 8.16% of subgroups.

Table 1 highlights subgroups where the ensemble model increased the most in terms of performance, (↑), maintained similar ones (=), or decreased (↓) performance relative to the individual models. Similar to the previous analysis, the subgroups here are also defined by combinations of planetary and stellar characteristics. To gain insight into how each individual parameter impacts the gain within these subgroups, we examined the Shapley values.

Figure 7 shows the Shapley values, which quantify the influence of each item. In the subgroups with the largest positive (Fig. 7(a)) and negative (Fig. 7(c)) gain when transitioning from the weak learner to the ensemble, the star's radius has a greater effect than the planet's surface gravity and the planet's semimajor axis, respectively. Interestingly, in the second subgroup, where individual and ensemble performance was unchanged, the planet's mass clearly contributes more than the stellar radius.

By looking at the Shapley values, we can thus see which factors matter the most for how much better or worse the ensemble model performs compared to the individual models in each subgroup. This helps us understand which characteristics about the planet or star make the most difference in how much the ensemble model helps or hurts predictions within different types of planetary systems.

| Subgroup | score | $\Delta_{score}$ | t |
|---|---|---|---|
| {*star_radius* $\in$ *[1.21, 1.35)* $R_{\odot}$, *planet_semi_major_axis* $\in$ *[0.04, 0.05)* $AU$} | 0.282 | -0.055 | 3.769 |
| {*planet_orbital_period* $\in$ *[2.78, 3.66)* $days$, *star_distance* $\in$ *[244.97, 348.43)* $pc$} | 0.284 | -0.053 | 2.651 |
| {*star_mass* $\in$ *[1.01, 1.07)* $M_{\odot}$, *planet_orbital_period* $\in$ *[4.64, 6.37)* $days$} | 0.287 | -0.051 | 2.447 |

Table 3: Top-3 highest positively divergent subgroups on performance for the *ensemble*.

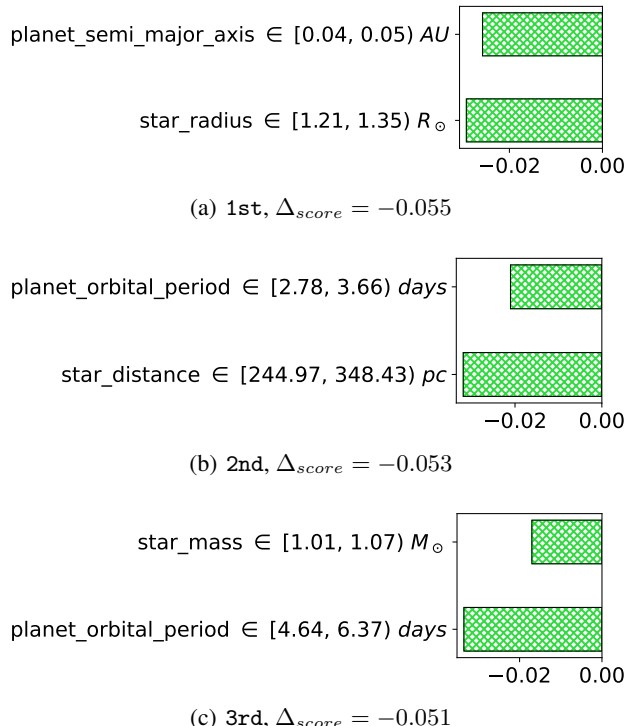

(a) 1st, $\Delta_{score} = -0.055$

(b) 2nd, $\Delta_{score} = -0.053$

(c) 3rd, $\Delta_{score} = -0.051$

Figure 6: ENSEMBLE. Item contribution to the final score for (a) the subgroup with the highest positive divergence (the lower the score, the better), (b) the second, and (c) the third.

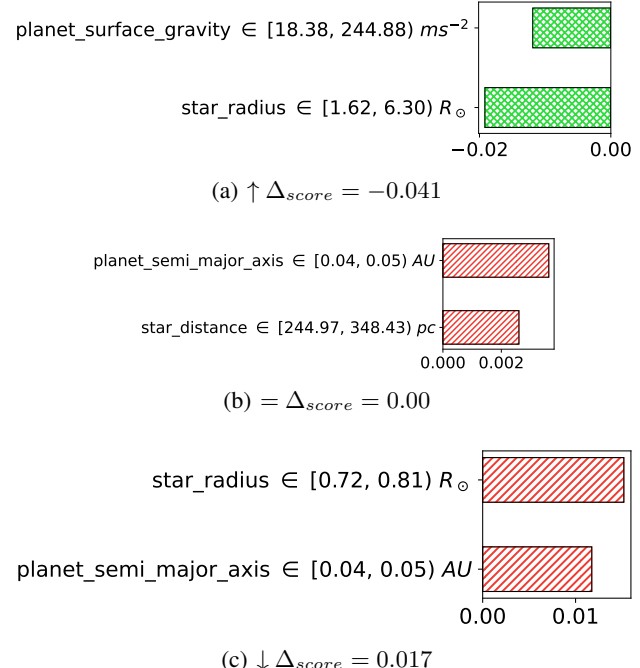

(a) $\uparrow \Delta_{score} = -0.041$

(b) $= \Delta_{score} = 0.00$

(c) $\downarrow \Delta_{score} = 0.017$

Figure 7: GAIN. Item contribution to the gain when comparing the ensemble with individual models. (a) the subgroup with the highest improvement when transitioning to the ensemble, (b) the subgroup for which the models perform equally, and (c) the subgroup with the highest decrease.

