# OpenReview forum: "Bad Exoplanet! Explaining Degraded Performance when Reconstructing Exoplanets Atmospheric Parameters"
_NeurIPS.cc/2023/Workshop/AI4Science — NeurIPS2023-AI4Science Poster_

### Official Review · Reviewer_ihTC · 2023-10-16
**Evaluation framework for exoplanets atmospheric parameter estimation**

**Rating:** 7
**Confidence:** 3

**Review:**

In this work, authors proposed an evaluation framework for exoplanet atmospheric parameter estimation models. By dissecting the dataset into multiple frequent subgroups, authors managed to identify specific subgroups where model prediction performance deteriorates. This framework will be helpful in recognizing subgroups where models underperform and assist with potential model (or training dataset) improvements.

Here are some specific comments regarding the framework:
1. As authors have pointed out, planetary metadata are continuous variables. In the experiments, certain bins are highlighted as worse-performing groups, and certain characteristics (e.g., high planet surface gravity) are assigned negative Shapley values (Fig. 4). It would be beneficial to look at these results in a continuous form: for example, plotting the array of Shapley values over the 10 bins drawn on planet surface gravity could provide hints on whether there exists a certain trend/hotspot.

2. Authors clearly showed that the ensemble of models outperformed individual weak learners. Instead of comparing an individual learner against the ensemble (such as Fig. 5), it would be more informative to compare different individual learners that composed the ensemble. Certain groups with high variance in performance might indicate an under-represented group that needs additional refinements or sampling.

3. The frequent pattern mining methods will find groups that compose at least 1% (or the specified threshold) of the dataset. It is kind of a dilemma here because deep learning models always tend to overfit the frequent groups while performing worse in less frequent groups. The most fatal error mode might fall in a minor group that composed less than 1% (or the specified threshold) of the dataset and will be missed by this evaluation framework. I wonder if there will be specific ways to compensate for such corner cases.

4. Authors mainly discussed model performances on simulation data. I am not familiar with how the ground truth labels are generated for the real exoplanets atmospheric parameters. Assuming they are scarce, it might be worth discussing how feedback from this evaluation framework can be organically incorporated into the development/refinement of an actual ML/DL prediction model, such as it can inform in which bin more ground truth labels are needed for further model training.